# PROTACs: Emerging Targeted Protein Degradation Approaches for Advanced Druggable Strategies

**DOI:** 10.3390/molecules28104014

**Published:** 2023-05-10

**Authors:** Nuwayo Ishimwe Sincere, Krishnan Anand, Sumel Ashique, Jing Yang, Chongge You

**Affiliations:** 1Laboratory Medicine Center, Lanzhou University Second Hospital, The Second Clinical Medical College of Lanzhou University, Lanzhou 730000, China; nuwayosincere@gmail.com (N.I.S.); ldyy_yangjingyj@lzu.edu.cn (J.Y.); 2Department of Chemical Pathology, School of Pathology, Faculty of Health Sciences, University of the Free State, Bloemfontein 9300, South Africa; krishnana1@ufs.ac.za; 3Department of Pharmaceutics, Bharat Institute of Technology (BIT), School of Pharmacy, Meerut 250103, India; ashiquesumel007@gmail.com

**Keywords:** PROTACs, proteolysis, emerging medicine, degradation, druggable technologies

## Abstract

A potential therapeutic strategy to treat conditions brought on by the aberrant production of a disease-causing protein is emerging for targeted protein breakdown using the PROTACs technology. Few medications now in use are tiny, component-based and utilize occupancy-driven pharmacology (MOA), which inhibits protein function for a short period of time to temporarily alter it. By utilizing an event-driven MOA, the proteolysis-targeting chimeras (PROTACs) technology introduces a revolutionary tactic. Small-molecule-based heterobifunctional PROTACs hijack the ubiquitin–proteasome system to trigger the degradation of the target protein. The main challenge PROTAC’s development facing now is to find potent, tissue- and cell-specific PROTAC compounds with favorable drug-likeness and standard safety measures. The ways to increase the efficacy and selectivity of PROTACs are the main focus of this review. In this review, we have highlighted the most important discoveries related to the degradation of proteins by PROTACs, new targeted approaches to boost proteolysis’ effectiveness and development, and promising future directions in medicine.

## 1. Introduction

The ubiquitin–proteasome system (UPS) is an essential part of the cellular machinery responsible for maintaining intracellular protein homeostasis [1,2]. A network of proteins that comprises the proteolytic system and chaperones calculates cellular protein homeostasis [3]. Chaperones are in charge of correcting protein misfolding, but the proteolytic system, which converges on the 26S proteasome, is in charge of removing damaged or unfolded proteins to maintain a healthy environment inside the cell. Using PROTAC technology for targeted protein degradation, a novel technique of treatment is emerging that stems from an aberrant expression of a protein that causes disease. PROTAC molecules are tiny, bifunctional molecules that bind an E3-ubiquitin ligase and a target protein at the same time, causing ubiquitination and proteasome destruction of the target protein.

Finally, they permit the production of an incredibly potent molecule with a higher margin of breakdown selectivity because of their catalytic property and the necessary ubiquitination process. The discovery of superior preclinical in vitro and in vivo PROTACs has sped up the development of therapeutically viable PROTACs [4]. A significant benefit of PROTACs is their intrinsic recycling; after dissociating from the target, the chimera can bind more free POIs and continue to iteratively promote their destruction. This event-driven process allows for the target to deteriorate in sub-stoichiometric quantities so long as the target contact is non-covalent [5]. With PROTACs (i.e., POI-specific warheads), lower concentrations than the parent inhibitor are possible because of their role as catalytic molecules. As a result, this might reduce negative side effects and increase the therapeutic window [6]. Another benefit of PROTACs over traditional pharmaceuticals is their capacity to successfully eliminate a target without an active site, which is made possible by the fact that the action mechanism (Figure 1) depends on the closeness of the E3 and POI [7]. This feature dramatically expands the proteome that can be drugged. Numerous studies also showed how chimeras with low-affinity ligands to their homologous POI might nonetheless achieve potent degradation [8,9] due to the advantageous cooperative interaction between the E3 and the target. As a result, PROTACs have a more profound and pervasive effect than parent molecule inhibition, significantly influencing the scaffolding function and downstream signal(s) of the target. Selectivity is still another characteristic [10].

## 2. Background, Composition, and Mechanism of Function

There has been evidence to support the hypothesis that some proteins can be ubiquitinated and degraded by the UPS by recruiting an E3 ligase in a way that is PROTAC-dependent ever since the discovery of PROTACs in 2001 [11,12]. Since its creation, PROTAC technology has found widespread application in both the development of novel therapeutics and fundamental biological studies. Over time, the application of the PROTAC technology has benefited a variety of distinct protein categories, including nuclear receptors [13], kinases, G protein-coupled receptors (GPCRs) [14], trans-membrane proteins, small GTPases, epigenetic proteins [15], transcription factors, and protein aggregates [16]. A linker connecting an E3 ubiquitin ligase (E3)-recruiting ligand and a protein gives a PROTAC molecule its dual function [17,18].

## 3. PROTACs Promote Targeted Cell or Tissue Selectivity

Given that PROTACs are successful in generating TPD, it is necessary to exploit this novel modality for therapeutic purposes [19,20]. The possibility of their use is however constrained by the risk of both on- and off-target toxicity [21,22]. The potential toxicities of PROTACs are projected to be less severe than those of traditional on-target inhibitors because they are event-driven technology. PROTACs can also be used in lesser doses due to their catalytic characteristics. Because they reduce the POI in areas far from the disease region, PROTACs have the capacity to significantly increase on-target toxicity. As an illustration, we consider how cancer tissues differ from normal tissues in their complete lack of selectivity for PROTAC-mediated proteolysis. To improve cell or tissue selectivity and lessen potential toxicity, a number of particular PROTACs have been developed. Targeted PROTACs have a number of promising applications, such as BCL-XL PROTACs, aptamer–PROTAC conjugates (APCs), hypoxia-activated PROTACs, folate-caged PROTACs, antibody–PROTAC conjugates (Ab-PROTACs), and photochemically controllable PROTACs (PHOTACs) [23,24]. The presence of certain elements, such as enzyme assistance, UV radiation, etc., causes the PROTAC molecule to become active. The process involved in the PROTAC molecule is shown in both active and inactive forms in Figure 2.

### 3.1. Photochemically Controllable PROTACs (PHOTACs)

By employing the photochemical modulation of PROTAC activity, PROTAC-mediated protein degradation can be spatiotemporally regulated, potentially preventing adverse effects [25,26]. Numerous studies and evaluations have been conducted on PROTACs that are photocaged and photo switchable [27]. Here, we demonstrate the effectiveness of this tactic using the examples of photocaged and photo switchable PROTACs.

#### 3.1.1. Photocaged PROTACs

In order to create photocaged PROTACs, either the POI ligand or the E3 ligase ligand can be caged, which results in an inactive degrader. The target protein will eventually degrade when the isolating group is taken out and the active PROTACs are released when exposed to light [28]. By connecting the 4,5-dimethoxy-2-nitrobenzyl group to the amide nitrogen of the JQ1 moiety, Xue et al. [28] produced a photocaged BRD4 decomposer in 2019. This is in accordance with a previous publication on the BRD4 degrader dBET1 [29]. According to a previous publication that identified dBET1 as the BRD4 degrader in 2019, Xue et al. [28] produced a photocaged BRD4 decomposer by adding the 4,5-dimethoxy-2-nitrobenzyl group to the amide nitrogen of the JQ1 moiety. However, PROTAC 4 only generated around 50% of the desired result.

#### 3.1.2. Photo Switchable PROTACs

Potential security concerns are raised when photocaged PROTACs, which are UV-sensitive, irreversibly release the active PROTAC molecules. Photo switching is a different approach to locally activate PROTACs (Figure 3). When Pfaff et al. (2019) joined the POI ligand and the E3 ligase ligand, they created photo switchable PROTACs using bistable ortho tetrafluoro azobenzenes (o-F4-azobenzenes) as a linker [30]. They chose ARV-771 [31], which has an approximate linker length of 11 between the POI ligand and the E3 ligase ligand, as the PROTAC lead structure. As shown in Figure 3, o-F4-azobenzene replaces the oligoether linker in ARV-771 to generate an isomeric pho-to-PROTAC pair, where the active degrader, trans-PROTAC 6, maintains the ideal spacing between the two ligands at 11 Å while the inactive degrader, cis-PROTAC 6, preserves the optimal spacing at 8 Å.

#### 3.1.3. Radiotherapy-Triggered Proteolysis

The systemic toxicity of PROTACs that could result from unexpected off-tissue protein breakdown might restrict their application in therapeutic settings. In order to precisely and spatiotemporally control protein breakdown by X-ray radiation, Yang et al. (2022) [32] announced the development of the irradiation-triggered PROTAC prodrug (RT-PROTAC activation technique). To create the first RT-PROTAC, we combined a PROTAC that targets the bromodomain (BRD) with an X-ray-inducible phenyl azide cage. X-rays can activate the RTPROTAC prodrug in vitro and in vivo even though it only has a marginal impact. In the MCF-7 xenograft model, activated RT-PROTAC exhibits synergistic antitumor activity with radiation and degrades BRD4 and BRD2 to the PROTAC degrader on an equal basis. Any potential non-specific toxicity issues should be addressed with PROTACs [33]. Figure 4 describes the radioactive PROTAC molecules in a tumor microenvironment.

### 3.2. Hypoxia-Activated PROTACs

Hypoxia is a common feature of solid tumors and is characterized by the overexpression of numerous proteins, including the epidermal growth factor receptor (EGFR) [34]. The hypoxic area of solid tumors has elevated nitro reductase (NTR) levels, enabling targeted prodrug activation in solid hypoxic tumors [27]). Similarly to how photocaged PROTACs are formed, hypoxia-activated PROTACs are created by introducing a hypoxia-activated leaving group (HALG) to the POI ligand or the E3 ligase ligand. Due to the increased NTR activity under hypoxic conditions, this design enables target protein degradation to occur after the active PROTACs are released. In 2021, Cheng et al. [35] produced a hypoxia-activated PROTAC by incorporating the 4-nitrobenzyl group into the EGFR ligand moiety of an EGFRDel19-targeting PROTAC. In HCC4006 cells, 58 PROTAC 7 showed a dramatically reduced affinity for the EGFR and were able to selectively degrade the EGFRDel19 in hypoxia rather than normoxia at a concentration of 50 M with a maximum degradation of 87%.

However, the modified PROTAC 8 could still effectively bind to EGFRDel19, and it was not clear whether the degradation caused by PROTAC 8 varies with normoxia and hypoxia. To create a new series of hypoxia-activated PROTAC, Shi et al. [36] most recently modified the VHL E3 ligase recruiter by adding the (1-methyl-2-nitro-1H-imidazol-5-yl) methyl group. Among the collected PROTAC molecules, PROTAC 9 might release the active PROTAC molecule in a hypoxic environment. Figure 5 depicts the hypoxia-PROTAC-molecule-based protein degradation.

### 3.3. Folate-Caged PROTACs

Although normal tissues have relatively low levels of this expression, the expression of the folate receptor 1 (FOLR1) is highly expressed in several human malignancies [37]. Due to this difference in expression, the conjugation of folate as a FOLR1 ligand offers possibilities for therapeutic targeting (Figure 6). The method of foliate conjugation has been successful in precisely delivering PROTACs to cancer cells. To create folate-caged PROTACs, a folate group is joined to the E3 ligase ligand of PROTACs using a cleavable linker [38]. The folate-conjugated PROTACs are transported inside cells by FOLR1, as shown in Figure 5, where the folate moiety is then degraded by endogenous hydrolases and reductase. The target protein has degradation-inducing active PROTAC results from this [39]. Based on the reported BRD4 PROTAC ARV-771, Liu et al. developed a cancer-selective PROTAC in 2021 [40]. When combined with the mother chemical ARV-771, the folate PROTAC 10 demonstrated comparable BRD4 degradation in cancer cells compared to non-cancerous normal cells. It was created by using an ester link to join folate to the hydroxyl group of ARV-771. BRD4 degradation is FOLR1-dependent, as shown by the fact that neither previously treated HeLa cells with free folic acid nor HeLa cells lacking endogenous FOLR1 had BRD4 decreased by the folate–PROTAC. The next year, they released another folate-caged PROTAC 11, which was made by joining the reported ALK with a folate group and a glutathione (GSH)-sensitive linker.

### 3.4. Antibody–PROTAC Conjugates (Ab-PROTACs)

Cytotoxic payload delivery to cancer cells can be targeted with the use of ADCs or antibody–drug conjugates. This makes it possible to have the most impact on cancer cells while having the fewest negative effects on healthy cells [40]. The creation of antibody–PROTAC conjugates (Ab-PROTAC) has been investigated as a novel method to increase the tissue and cell-type selectivity of PROTACs on the basis of the ADC concept. By affixing a tumor-cell-specific antibody to a PROTAC molecule using a cleavable linker, Ab-PROTACs are created [41]. Following internalization of the antibody PROTAC, the linker of the Ab-PROTACs can be hydrolyzed, releasing the active PROTAC molecule. The antibody binds exclusively to tumor cells; this is analogous to how folate-caged PROTACs function.

### 3.5. Conjugates of Aptamer–PROTACs (APCs)

Aptamers, sometimes referred to as single-standard nucleic acids, are highly selective and affine when they bind to their intended protein targets [42]. Aptamers are commonly used in targeted therapy for human malignancies because of their small physical size, flexible structure, quick chemical manufacturing, varied chemical modification, excellent stability, and lack of immunogenicity [43]. In order to improve the capacity of traditional PROTACs to target particular tumors and their anticancer efficiency in vivo, he and his colleagues developed the conjugation of aptamer and PROTAC (APC) method in 2021 [44]. After being specially recognized by tumor cells that overexpress nucleolin, GSH breaks the disulfide bond, releasing the active PROTAC molecule.

### 3.6. Minimal Platelet Toxicity with BCL-XL PROTACs

B-cell lymphoma extra-large (BCL-XL), a well-validated therapeutic target for cancer, is largely overexpressed in a variety of solid tumor cells as well as in a subset of leukemic cells [45,46]. The on-target and dose-limiting thrombocytopenia caused by BCL-XL inhibition substantially restricts the use of BCL-XL inhibitors in medicine [47,48]. To reduce platelet toxicity, Khan et al. in 2019 developed a selective navitoclax-based BCLXL PROTAC degrader that directs BCL-XL to the minimally expressed VHL E3 ligase [49]. MOLT-4 T-cells from acute lymphoblastic leukemia may produce PROTAC 14, which could break down BCL-XL.

## 4. Past, Present, and Future Advancement

### 4.1. The History of PROTAC (2001–2016)

A rundown of PROTAC’s development milestones and a list of the proteins that it has successfully controlled is being under full consideration at this current time. The first PROTAC was released by the Crews and Deshaies laboratory in 2001, and it utilized the SCFb-TRCP E3 to inhibit methionine ami-nopeptidase-2 (MetAp-2) [12]. Following this original discovery, it was discovered that PROTACs induce the degradation of the androgen (AR) and estrogen (ER) receptor, expanding the target spectrum. The function of these AR- and ER-targeting PROTACs in an intact cell was shown by microinjection. The subsequent PROTACs used the Von Hippel indau disease tumor-suppressor protein (VHL) E3 to bind to intact cells without the need for microinjection by using a portion of the HIF-1a peptide with a cell-penetrating peptide sequence [49]. Later, a PROTAC containing a shorter HIF-1a peptide fragment was created. The first PROTAC-recruiting cIAP1 was created by the Hashimoto group, who also employed similar bestatins to specifically target and eliminate cellular retinol- and retinoic-acid-binding proteins (CRABP-I and II) [50,51,52,53]. Later, in 2012, researchers developed peptidomimetic ligands that have a high affinity for VHL E3 [54]. Later, the Ciulli laboratory reported on further structure–activity relationship (SAR) investigations of the VHL ligand that produced VHL ligands with superior physical chemistry properties but the same affinity to VHL. Researchers discovered that the E3 cereblon is the main molecular target of the immunomodulatory medicines (IMiDs) thalidomide, pomalidomide, and lenalidomide throughout the same time period. IMiDs have been observed to promote the recruitment of neosubstrates for ubiquitination and proteasomal degradation, such as Ikaros, as soon as they bind to CRBN [55]. Figure 7 describes the pipeline of PROTACs from 2001 to future aspect.

### 4.2. Current: Target Scope for the PROTAC

Transmembrane RTKs and a modern example of a lipid kinase, phosphatidylinositol 3-kinase (PI3K), which is among the most researched PROTAC targets [57], are found in the BET [58] and kinase [59] families. The p300/CBP-associated factor (PCAF), general control nonderepressible 5 (Sirtuins, Sirt2), other protein families, histone deacetylases (HDACs), Sirtuins, Sirt2, and other epigenetic targets (GCN5) are other recent cases. Over the past year, homo-PROTACs have been created that automatically target VHL and CRBN [60]. Despite the fact that the majority of PROTACs use small molecules as their E3 recruitment moiety, there have been recent examples of PROTACs that target proteins using the HIF-1a peptide [61]. In an effort to comprehend the mechanism of PROTACs better, the development of the ternary complex (POI-PROTAC-E3) has recently received considerable attention due to their catalytic abilities. PROTACs have been referred to as “programmable essential activators” in the context of ubiquitin ligase enzymes [62]. In addition to being necessary for ubiquitination transfer and programmable, since they can be programmed to target any POI, PROTACs also catalytically mediate the formation of ternary complexes, making them operate as activators. Consequently, considering PROTACs as activators can serve as a basis for more efficient PROTAC design. In conclusion, when developing PROTACs, it is critical to consider how ternary complexes evolve. Based on the information provided, it is possible for both binary POI-ligand affinity and PROTAC affinity to exist. Recent research indicates that the mechanism of action (MOA) of PROTACs for degradation can enhance selectivity among homologous targets when compared to inhibition. Specifically, two lines of evidence from the study suggest that PROTACs may be more effective at achieving selectivity than traditional inhibitors [9].

Both studies employed the highly homologous kinase protein family for their investigation. In PROTAC, the location of the linker attachment and the length of the linker can also affect selectivity and degradation patterns. According to a recent study, the PROTAC linker was necessary for the ternary complex structure of CRBN and BRD4. This demonstrates that a PROTAC with the same E3-recruiting ligand and POI ligand can display different selectivity patterns depending on changes in the linker attachment sites and chemical makeup. The degradation profiles are also impacted by the linker length. For instance, when the linker length is raised by three atoms, a lapatinib-based PROTAC degradation profile switches from targeting both EGFR and HER2 to only degrading EGFR. Based on a study from the previous year, it has been suggested that the event-driven PROTAC (Proteolysis-Targeting Chimeras) mechanism of action may be effective in preventing inhibitor-resistance mechanisms such as target protein overexpression and mutations [63]. A common resistance mechanism in response to therapeutic inhibition is target protein mutation, such as changes close to the inhibitor binding site that render treatment inefficient or ineffective. This is evident, for example, when the medication ibrutinib is used to treat CLL [64], which frequently depends on BTK activity. PROTACs can also defeat other inhibition-derived resistance mechanisms. An AR inhibitor known as is used to treat prostate cancer. Enzalutamide-induced resistance mechanisms include endogenous androgen ligand upregulation and/or AR mutations that become agonists. Scaffolding and enzymatic roles can be altered by targeting kinases with PROTACs; this is in contrast to occupancy-driven pharmacology, which simply affects only enzymatic functions. In a recent study, it was shown that PROTACs targeting RTKs could target frequently occurring RTK mutations that are resistant to kinase inhibitors and are degraded by scaffolding activities. Furthermore, kinome rewiring was delayed by PROTAC-induced degradation relative to inhibition. These drugs’ capacity to successfully induce the breakdown of a number of AR mutants for which inhibition is no longer active demonstrates their event-driven nature. Further research revealed that when synthetic androgen (R1881) was added, these PROTACs outperformed enzalutamide in preventing the proliferation of AR overexpressing cells and promoting death [65].

Additionally, for the first time, this illustration demonstrated how PROTACs can result in the degradation of transmembrane receptors. It has also been demonstrated that the focal adhesion kinase (Fak) scaffolding activities are impacted by the PROTAC-induced degradation of Fak [66]. Fak signaling, cell migration, and invasion in triple-negative human breast cancer cells were found to be improved by induced degradation over kinase activity suppression. The expression of a number of inflammatory mediators in LPS-activated macrophages and dendritic cells was also potently modified by a PROTAC, of which targeted the closely related epigenetic proteins PCAF and GCN5. The POI ligand, on the other hand, was unable to change the immunomodulatory activity of PCAF and GCN5 when used to block their bromodomains.

## 5. Future Perspectives: Developing PROTAC Technology and Expedite PROTAC Discovery

It is crucial to identify the guiding principles and provide reliable evaluation frameworks [67]. The POI recruitment moiety in the majority of reported PROTACs to date is a well-studied ligand, often an inhibitor. For instance, the PROTAC design is influenced by the POI ligand’s crystal structure and the POI ligand’s structure–activity relationship (SAR) data. In a recent example, the putative transcription factor regulator was selectively targeted using PROTACs that were designed using the physicochemical properties. In addition, end-point methods, such as mass spectrometry and, more recently, immunoblotting, are commonly used to evaluate the activity of PROTACs. These techniques, however, do not yet work well for high-throughput screening, and they provide little guidance for PROTAC structural optimization. A ternary complex structure and binding data would be useful sources of information for determining PROTAC efficacy. There are currently not many known crystal forms of a PROTAC-mediated ternary complex. There have only been a few crystal formations described thus far. For PROTAC evaluation, Promega has developed a real-time live-cell method that enables both the characterization of effectiveness and MOA [68]. The technique uses light technologies and endogenous CRISPR/Cas9 tagging to kinetically assess target protein levels in living cells. Selecting a target is one of the most important decisions in a drug discovery effort [69].

As previously stated, the process of developing a PROTAC (PROteolysis-TArgeting Chimeras) can be time-consuming, and it is possible that not all biological targets may be suitable for protein breakdown using this approach. Therefore, it would be quite advantageous to receive early confirmation of the POI objectives. It was shown that several therapeutic targets for kinases were vulnerable to degradation brought on by PROTAC. Similar investigations may be beneficial for other protein families prior to beginning PROTAC’s development. Additionally, protein labeling techniques such as HaloTag (HT) [70] and His-tagged [71] POI can be used to detect the target’s degradability, which aids in target validation. The vast majority of the target proteins developed so far for PROTACs are still found in the druggable proteome. It might be conceivable to develop compounds that can modulate challenging non-traditional pharmacological therapeutic targets with the use of PROTAC technology. The PROTAC technique only requires binders that provisionally mediate ternary compound formation, enabling the inclusion of low affinity POI ligands. By the end of 2021, 15 PROTACs (Table 1) will have started clinical trials [72]. Figure 8 depicts the role and future pipeline of PROTAC molecule in drug discovery.

## 6. Turning PROTAC into Medications

All molecules are enormous by the standards of traditional tiny molecules since it is advantageous to combine all three components into a single molecular unit, which is typically 300–500 Da. PROTACs typically have a size between 700 and 1000 Da. Due to their larger size, they are more challenging to give orally, yet once in the bloodstream, they behave like typical small molecules. This means that these molecules behave as expected for tiny molecules when given intravenously, intraperitoneally, or subcutaneously: exposures are inversely correlated with dose, molecules are well distributed throughout the body, and chemicals that have been optimized have little liver clearance. It is particularly noteworthy that numerous studies have shown that target proteins are vigorously degraded in vivo during preclinical research. For instance, subcutaneous Brd4 PROTAC administration has been shown to cause >90% Brd4 breakdown in tumor tissue [27]. As a result, many issues related to making PROTACs accessible can be handled using standard medicinal chemistry methods. Due to PROTACs’ exceptional catalytic abilities, target degradation and the related efficacy in preclinical models can frequently be shown at plasma concentrations as low as 100–200 nanomolar. As a result, the choice of distribution technique is greatly influenced by the selective protein and illness indication. If the efficacy may be achieved by intermittent dose, such as by causing tumor cell death or removing a toxic, aggregated protein, then administering a PROTAC intravenously is the quickest way to the clinic. The gastrointestinal absorption phase necessary for oral drugs is avoided with intravenous delivery, and a wide variety of well-known prescription formulations are compatible with PROTACs. Additionally, a relatively low dose is frequently anticipated for therapy success due to PROTACs’ unusually high potency. Although patients typically do not pick IV ports for drug delivery, more recent drug delivery systems can permit continuous IV dosage. The subcutaneous approach has generated a lot of interest recently. The proteasome inhibitor bortezomib (Velcade), which is typically administered intravenously, can now be administered subcutaneously thanks to a bridging study that showed it to be just as effective.

Both antisense and protein treatments have been successfully employed in conjunction with this strategy to deliver payloads [73]. Subcutaneous drug administration may result in a depot effect, which reduces the need for frequent dosing. The SC method is preferred by patients over the IV method; however, there are two issues that need to be resolved. Firstly, only extremely small volumes—typically much less than 1 mL—are allowed for SC injections. The desired dose must therefore be soluble in this volume. Secondly, the subcutaneous region frequently experiences issues with injection site tolerability. For instance, commonly used formulation ingredients might result in SC pain and injection site granulomas even for medicines that have been approved [74]. For subcutaneous distribution, a PROTAC must therefore have great solubility in a safe formulation.

This strategy has several drawbacks, but it has the advantage of allowing for PROTACs to remain continuously exposed for a considerable amount of time while also being readily assimilated into the core compartment. New delivery techniques can also greatly enhance the injection volume, making it easier to distribute PROTACs that cannot be produced in minute quantities. Oral delivery is still the favored method for tiny compounds. This administration strategy, which also preserves medication exposure, is the simplest and most common. Additionally, oral medications may employ a number of legal formulations, allowing for the distribution of chemical substances that are not particularly soluble. The absorption of oral PROTACs may be increased theoretically by including permeability enhancers and efflux inhibitors in the drug formulation. On the negative side, oral drugs make patient monitoring more challenging. The oral bioavailability of PROTACs is currently being improved through a number of projects. Finally, it may be more challenging to determine the right dose regimes for PROTACs than for conventional inhibitors since too much of a PROTAC may lead to the production of two dimers rather than a trimeric complex. If either the target protein or the E3 ligase is greatly overrepresented in the PROTAC stoichiometry, the ternary complex cannot form. The two distinct PROTAC molecules will bind the two proteins. Therefore, if the stoichiometry of the PROTAC is much higher than that of the target protein or an E3 ligase, the formation of the ternary complex is precluded. This worry has been somewhat reduced by the direct in vitro and in vivo observation of deterioration. Several data reveal substantial evidence that cereblon-based PROTACs are superior to conventional small-molecule inhibitors in their ability to target BRD4. Findings support convincing evidence that cereblon-based PROTACs target BRD4 more effectively and more effectively than conventional small-molecule inhibitors. These results thoroughly confirm the design of BRD4 PROTACs as an innovative and promising method to effectively target BRD4. Additionally, this study is the first to outline an effective PROTAC that functions by enlisting the E3 ligase cereblon. Their research shows that using the PROTAC platform to target the E3 ubiquitin ligase cereblon or other E3 ligases has considerable promise for developing potent treatments [75].

As an illustration, in vitro tests using Brd4 PROTAC demonstrate that Brd4 completely degrades between 1 nM and 300 nM, with only a little reduction in degradation observed at 1 M [76]. The in vivo studies [27] were unable to demonstrate whether a higher dose prevents Brd4 from degrading between 1 mg/kg and 10 mg/kg in a dose-proportional manner.

## 7. PROTACs: Targeted Management Strategy for Several Diseased Conditions

### 7.1. PROTACs in the Cancer Treatment

The development of better cancer treatments is a priority for the medical community. Recent years have seen the development of innovative cancer therapeutic approaches, including CRISPR/Cas9 technology, RNA interference methods, antisense oligonucleotides, monoclonal antibodies, and small-molecule inhibitors (SMIs). CRISPR/Cas9 technology can only be used as a candidate technique, though, because the off-target effect is still a problem. Only a limited portion of ASOs now have FDA marketing approval [58] due to a number of difficulties with RNA interference techniques and ASO medicines, including biological instability, immunogenicity, cell transport, and biological distribution [77]. Antigen-specific binding, which reduces cell toxicity, is the idea behind monoclonal antibodies. However, their inability to target intracellular proteins or receptors along with their high cost precludes their application. Antigen-specific binding is the idea of monoclonal antibodies, which helps reduce cell toxicity. They cannot, however, target intracellular proteins or receptors, and their high cost precludes their use in medicine [48].

Despite the present widespread use of small-molecule modulators that target tumor cells, there are still several downsides, including drug resistance, toxicity, and poor selectivity. Finding new treatment approaches is essential to overcoming the downsides of current drugs. The development of PROTAC has generated a lot of anxiety [78]. This approach effectively avoids the problem of medication resistance by selectively destroying pathogenic proteins via the UPS, a unique intracellular protein degradation mechanism. Recently, various research groups have focused on PROTAC technology and have produced a large number of small PROTAC compounds for a variety of malignancies, including many cancers [44]. The “De-Linker” unique graph-based deep generative model, which combines 3D structural data with cutting-edge machine learning techniques for scaffold hopping and fragment linking, was recently published by certain teams. As a result, as seen in various examples, it can also be used while designing PROTACs. Additionally, we believe that in the future, the application of computational generative approaches in drug discovery will increase [79]. A few PROTAC compounds’ IC_50_ values can be as low as 1 nM in certain contexts.

In addition to its superior ability to break down proteins, PROTAC has the ability to accelerate tumor regression, decrease tumor development, and create advanced anticancer effects on the cellular level. The mechanism of PROTAC against cancer cells is shown in Figure 9.

In neurodegenerative diseases, which can worsen over time and lead to malfunction, neurons or their myelin sheaths are destroyed. Cerebellar atrophy, Parkinson’s disease, Huntington’s disease, and Alzheimer’s disease (AD) make up the majority of them clinically. Alzheimer’s disease (AD) is one of the most common neurological disorder and its clinical symptoms mostly include cognitive decline, behavioral abnormalities, and a reduction in functional capacity [80]. The causes of AD are complex and comprise several variables. The pathogenic tau protein and the -amyloid (A) cascade are among those that have been intensively studied; however, recent findings suggest that the latter is more likely to be the focus of treatment for AD [75], given that tau proteins are degraded by the peptide-based PROTAC TH006 [60]. An E3 ligand and a tau were the primary additions to the fusion peptide molecule. PROTACs, which stand for PROteolysis TArgeting Chimeras, have emerged as a promising therapeutic approach since their discovery in 2001. One of the main advantages of PROTACs is their ability to target and degrade proteins that were previously considered “undruggable,” making them a potentially important tool in treating a variety of diseases. In particular, in vitro studies have shown promise for using PROTACs as a therapeutic modality in triple-negative breast cancer (TNBC). Although the data suggests great potential, further research is needed, including in vitro and in vivo clinical trials, to fully establish the safety and efficacy of using PROTACs in TNBC treatment [81]. PROTACs have emerged as a promising class of therapeutics for lung cancer and drug resistance in recent years. Researchers have developed several PROTAC molecules targeting validated therapeutic targets in NSCLC such as EGFR, KRAS, ALK, BRAF, and BCL-XL. These compounds have shown antitumor efficacy in cell models and preclinical tumor models. The development of PROTACs is rapidly advancing, and they hold great potential as novel therapies for lung cancer [82]. PROTACs have been the subject of research for the past two decades, but only a few have demonstrated selectivity towards tumor cells. This is because many PROTACs recruit E3 ligases that are expressed ubiquitously in both normal and tumor tissues, which can lead to on-target toxicities. However, researchers are exploring various strategies to enhance the selectivity of PROTACs for tumor-specific proteins of interest (POIs) to improve their therapeutic potential [83].

### 7.2. PROTACs in Immune System Diseases

Autoimmune disorders develop when the body’s immune response to self-antigens damages its own tissues. They can be added to by using acquired immunological disorders and are characterized by different levels of molecular and tissue damage, even organ failure and death [84]. Genes, the environment, and risky habits such as smoking all have a role in the etiology of autoimmune illnesses. Despite extensive study and significant progress, we still need to completely understand the mechanism and make additional efforts [85]. IRAK4 activation has been associated with autoimmune diseases such SLE, psoriasis, rheumatoid arthritis, and cancer [86]. As a result, Harling and his colleagues recently published a PROTAC molecule to target and destroy IRAK4 [87]. The authors of this work successfully predicted the ideal role to bind IRAK4 ligands using molecular docking analysis based on the crystal structure of IRAK4 along with its kinase inhibitors. They then conducted a search on the constructed model and identified 12 atoms to be an appropriate duration of linker. General control non-derepressible 5 (GCN5) and P300/CBP-associated factor (PCAF), two related epigenetic proteins, can lower immune factors such as IL-6 and TNF that are caused by genes, the environment, and unhealthy habits such as smoking [88].

We still need to completely understand the mechanism and make more effort despite extensive research and significant advancements [85].

### 7.3. PROTACs in Viral Infection

The hepatitis B virus (HBV) mostly causes acute and chronic hepatitis B. The health of the entire world’s population is already at risk from HBV because millions of people are affected [89]. These HBV-infected patients will eventually develop mild or light hepatitis, liver cirrhosis, and liver fibrosis, which will progress to hepatocellular cancer. Therefore, it is believed that HBV is the main cause of HCC [90]. The two primary drugs used to treat HBV infection nowadays are reverse transcriptase inhibitors and interferons. However, neither of them makes a major difference in the fight against HBV. To fully address the issue of HBV infection, there is still considerable effort to be carried out. The X-protein trans-activator of HBV controls apoptosis, transcriptional networks, DNA repair networks, and intracellular signal transduction pathways. The X-protein trans-activator of HBV controls apoptosis, transcriptional networks, DNA repair networks, and intracellular signal transduction pathways [91].

Studies have shown that the pathogenesis of HCC caused by HBV depends on the X-protein. It is clear that the X-protein promotes intracellular transcription and replication of HBV while blocking hepatic molecular apoptosis, hence promoting the growth of liver cancer cells. Apoptosis and HBV transactivation may be prevented by the X-protein’s ubiquitin-binding, unstable, and oligomerization regions, according to a previous publication. In 2014, a number of peptide-based PROTAC compounds with an emphasis on the X-protein were developed [92].

The N-terminal oligomerization area and the C-terminal instability region of PROTAC molecules were joined by fusion of the X-N-proteins and C-terminal unstable sections, resulting in the formation of two distinct types of PROTAC molecules. Additionally, they were able to create several fusion peptides by replacing the C-terminal dangerous region with the oxygen-structured degradation (ODD) region of hypoxia-inducible factor (HIF-1), which interacts with VHL-type E3 ligase [93]. Due to the previous ligand, they had been equally effective at destroying PROTAC, proving that the C-terminal’s dangerous area can also be utilized as an E3 ligase ligand. The potential for the PROTAC compounds to prevent or treat HBV infection and/or HCC should be investigated in preclinical studies.

### 7.4. PROTACs in Metabolic Disease

Molecular docking technology, fortunately, provides a speedy and efficient means to complete jobs and can greatly reduce prior preparation labor. A useful resource is the ability of molecular docking techniques to automatically discover putative binding sites between targets and ligands for some unidentified proteins [94]. PNPLA3 (patatin-like phospholipase domain-containing protein 3) and prenyl-protein chaperone PDE are two examples of studies devoted to learning more about particular unknown proteins [95], and PROTAC molecules have the capacity to degrade those target proteins. PROTAC technology, on the other hand, is a state-of-the-art method that can assist in obtaining profound insights into the mechanisms and functions of unknown proteins. Numerous studies have shown that PROTAC technology has the ability to continually influence with advanced discoveries that enhance the science of drug discovery.

## 8. Outlook for the Next 20 Years of Targeted Protein Degradation (TPD)

Beyond the current list of medications in clinical testing and those in various firms’ development pipelines, where should the TPD field aim to reach in the next 20 years? What opportunities and issues still exist in the sector, and what could be accomplished if those were looked into and resolved? Which novel therapies, such as E3 ligases, ligands, and certain protein degrader classes, would be most helpful for which disorders? The following milestones in this “new era” of TPD, in our opinion, will focus on four clinical translation inflexion points: defining and clinically demonstrating the target classes best served by degradation over inhibition; expanding the clinical reach of the modality outside of oncology; and validating TPD modalities other than IMiDs and PROTACs in clinical settings [96,97].

## 9. Targeted Protein Degradation: LYTAC and AUTAC

TPD, or targeted protein degradation, has gained popularity as a method for creating novel drugs to cure illnesses. Other TPD approaches are emerging in addition to the extensively researched PROTAC method, including molecular glue, the lysosome-targeting chimera (LYTAC), the antibody-based PROTAC (AbTAC), and the autophagy-targeting chimera (AUTAC). By offering new perspectives, these novel approaches have increased the possibilities for TPD and are advancing drug discoveries [98].

### 9.1. Lysosome-Targeting Chimaera (LYTAC)

Through the endosome–lysosome route, LYTAC is a unique method to promote the degradation of extracellular and membranous proteins [99,100] (Figure 10). Since extracellular and membrane proteins make up a sizeable fraction of encoded proteins and can be extremely important in conditions such as cancer, autoimmune illnesses, and neurodegenerative diseases, LYTAC is possibly a useful addition to PROTACs. A ternary complex is created when LYTAC molecules bind an extracellular or membrane protein of interest (POI) and a lysosome-targeting receptor (TLR) that is present on the cell surface. Through clathrin-mediated endocytosis, this complex then causes the uptake of POI and eventual degradation. The first LYTAC molecule ever described made use of the IGF2R, also known as the cation-independent mannose-6-phosphate receptor (CI-MPR). CI-MPRs (cation-independent mannose-6-phosphate receptors) are essential for the movement of lysosomal enzymes inside cells. N-glycans bound with mannose-6-phosphate (M6P), which are identified by CI-MPRs, modify these enzymes. Because of the low pH in late endosomes, lysosomal enzymes separate from the CI-MPRs. Following this, the enzymes are directed toward lysosomal destruction, while the CI-MPRs are carried to the Golgi apparatus and the cell surface for recycling [101]. The use of LYTAC molecules is a cutting-edge therapeutic approach that targets protein degradation through a natural process. In this method, LYTAC molecules are produced by combining a small molecule or antibody with a synthetic ligand that targets the CI-MPR poly-M6Pn.

The ability of LYTACs to degrade numerous therapeutically important proteins has been proven. For instance, scientists have successfully employed a LYTAC molecule to selectively degrade EGFR in a number of cell lines by covalently conjugating poly-M6Pn to the EGFR antibody, cetuximab. According to a different investigation, conjugating poly-M6Pn with an anti-PD-L1 antibody significantly reduced the level of PD-L1 expression on the cell surface [99]. While other LTRs are exclusively expressed in a few tissues, CI-MPR is distributed widely throughout the body. It could be able to selectively induce the breakdown of target proteins in certain tissues by focusing on tissue-specific LTRs. One liver-specific LTR is the asialoglycoprotein receptor (ASGPR) [102,103]. Antibodies and N-acetylgalactosamine (GalNAc), which particularly recognize ASGPR, are combined to generate the LYTAC molecule that targets ASGPR. The effectiveness of this method in targeting cells that express ASGPR has been proven by co-culture studies. Further research into alternative LYTACs is warranted given the initial success of the CI-MPR- and ASGPR-based LYTACs [100].

### 9.2. Autophagy-Targeting Chimera (AUTAC)

A second pathway, in addition to the endosome–lysosome system, for targeted protein degradation is the autophagy lysosome pathway [104,105]. An essential role in the cell’s recruitment of autophagosomes is played by the signaling molecule 8-nitro-cGMP. This is a modified form of cyclic guanosine monophosphate (cGMP) [106]. Autophagy-targeting chimeras (AUTAC) have been created using 8-nitro-cGMP, as shown in Figure 11. AUTAC consists of three parts: a cGMP-based degradation tag, a linker, and a small-molecule ligand that binds to a particular protein or organelle of interest (POI) [104]. There are two categories of chemicals that, by various processes, cause the breakdown of proteins in cells: K63-linked polyubiquitination and lysosome-mediated degradation are induced by AUTAC molecules, whereas K48-linked polyubiquitination and proteasome-mediated degradation are induced by PROTAC molecules.

AUTAC4, a new AUTAC molecule described in a recent study by Takahashi et al., encourages the mitophagy of fragmented mitochondria. AUTAC4 removes damaged mitochondria by attaching to a transporter on the outer mitochondrial membrane, utilizing a derivative of 2-phenylindole. Treatment with AUTAC4 was reported to efficiently promote mitophagy in cells with fragmented mitochondria and restore mitochondrial membrane potential and ATP production [104]. The findings imply that AUTAC has a broad range of uses, and future research into AUTAC in a number of scenarios, such as protein aggregation breakup, is anticipated.

### 9.3. GlueTAC

Another lysosome-based method to degrade cell-surface proteins, known as GlueTAC, has recently been created [107] (Figure 9). To speed up degradation, GlueTAC uses Nano bodies, covalent bonds, a cell-penetrating peptide, and a lysosome-sorting sequence (CPP-LSS). Covalent interactions boost binding affinity and minimize off-target effects, while CPP-LSS promotes internalization and lysosomal breakdown [108,109]. Nanobodies also substitute traditional antibodies for cell penetration. The creation of a GlueTAC chemical that targets PD-L1 served as proof of GlueTAC’s potency. An FDA-approved PD-L1 antibody, atezolizumab, was found to be less effective than this chemical in lowering PD-L1 levels in cells and preventing tumor growth in immunodeficient mice. Although GlueTAC is an innovative new method for breaking down cell-surface proteins, there are several issues that must be taken into account. First, as GlueTAC molecules add unnatural amino acids into nanobodies and form covalent interactions between nanobodies and antigens, their safety needs to be carefully considered. Second, it is crucial to remember that nanobodies lack heavy chains and, as a result, cannot bind to FcRn. This may cause GlueTAC to have a shorter half-life than antibodies [110,111]. The period of time it takes for half of a substance to break down or decay is known as the half-life. You can conduct experiments in which you measure the amount of a substance at various points following its synthesis or introduction into a system in order to calculate the half-life of the substance.

### 9.4. Antibody-Based PROTAC (AbTAC)

AbTACs, a novel class of targeted protein degraders, utilize the specificity of antibodies to trigger the degradation of intracellular and membrane proteins. These protein degraders are designed as antibody-based PROteolysis-TArgeting Chimeras (AbTACs) and can selectively target specific proteins for degradation. AbTACs have the ability to recognize and bind to specific antigens, making them a promising approach for targeted protein degradation. AbTACs use an antibody to specifically bind to a target protein and bring it into contact with an E3 ligase, which causes ubiquitination and eventual degradation. This is in contrast to regular PROTACs, which normally rely on small molecules to recruit E3 ligases to target proteins. Although AbTACs were first designed for intracellular targets, more recent research has shown that they can also be used to cause the degradation of extracellular and membrane proteins [112]. By utilizing the lysosome pathway, AbTAC technology has the potential to significantly increase the number of proteins that may be targeted for destruction. Using bispecific antibodies, one arm of which is directed against a cell surface protein of interest (POI), the other arm is directed against a transmembrane E3 ligase, such as RNF43. The internalization of the POI and subsequent destruction in the lysosome are caused by the addition of the AbTAC molecule. While LYTAC, which similarly induces the TPD of cell-surface POI, and AbTAC have similarities, the latter’s mechanism of action is less understood. For instance, it is not yet known if the intracellular portion of the POI is ubiquitinated before endocytosis and, if so, how this ubiquitination affects internalization.

## 10. Therapeutic Application of PROTAC

Proteolysis-targeting chimeras (PROTACs) have fundamentally altered the way that medications are created due to their numerous advantages over traditional small-molecule inhibitors. As opposed to occupancy-driven inhibitors, PROTACs’ mechanism of action (MOA) is event-driven and catalytic in nature, leading to a stronger and longer-lasting effect. Furthermore, PROTACs offer a further level of selectivity that reduces potential toxicity and boosts effectiveness in the face of drug-resistance mechanisms. By concentrating on non-enzymatic processes, they can increase the range of potential treatment targets.

Since its discovery twenty years ago, PROTACs have developed from cell-impermeable peptide–small-molecule hybrids to clinical candidates that are orally bioavailable and can break down oncogenic proteins in people. The pace of scientific advancement is expected to quicken as we approach the third decade of targeted protein degradation (TPD). The creation of ligands for previously “undruggable” proteins and the recruitment of new E3 ligases are made possible by advances in technology. Furthermore, improved computational power is assisting in the logical design of more powerful and selective PROTACs as well as the discovery of active degraders [113]. When opposed to conventional inhibitors, PROTACs, also known as proteolysis-targeting chimeras, offer a novel pharmacodynamic strategy with a number of potential advantages. The ability to achieve pharmacodynamic efficacy even when the PROTACs are not detectable in the body is one of these benefits [114]. Selectivity, effective distribution, and sensitivity to drug resistance are just a few of the advantageous characteristics of PROTACs. These features can be enhanced by using targeting ligand methods. These compounds function by selectively stimulating intracellular proteolysis, which has been shown to be effective in inhibiting cancer cell proliferation and encouraging apoptosis [115]. A particular class of molecule known as PROTAC can benefit from the advantageous interaction between two amyloid proteins. The PROTAC-induced close proximity of the amyloids allows for the formation of a stable ternary complex, which can boost the cross-interaction’s beneficial effects. The PROTAC molecules will be built using peptide mimics with a high affinity for a 1-42 [116,117].

Alzheimer’s disease and related tauopathies are characterized by tau buildup within cells, and targeting tau has emerged as a promising strategy for therapeutic development. In order to selectively degrade proteins inside of cells, the proteolysis-targeting chimera (PROTAC) approach was developed. In order to achieve this, a novel small-molecule PROTAC with the designation C004019 and a molecular mass of 1035.29 dalton was developed. PROTAC was created to specifically increase tau protein ubiquitination and proteolysis by recruiting tau and E3-ligase (VHL) [118]. The first VHL-based small-molecule PROTAC designed to target nuclear hormone receptors was called PROTAC-ERR. These PROTACs have a lot of medicinal promise [119]. A substance called PROTAC-ERR can aim to degrade estrogen-related receptor alpha. The orphan receptor estrogen-related receptor alpha (ERR), which is found in the nucleus of MCF-7 breast cancer cells, can be targeted for degradation by the substance PROTAC-ERR.

At a dose of 100 nM, PROTAC-ERR can cause these cells to degrade approximately 50% of ERR. The tyrosine kinase family of enzymes, which JAKs are a member of, is involved in the transmission of cytokine-mediated signals in cells. JAKs phosphorylate tyrosine residues when they are activated, which can subsequently activate downstream signaling proteins and cause a variety of physiological effects. These enzymes are able to transmit information from external chemicals such as cytokines, growth factors, and chemokines to the cell’s nucleus, where they can directly affect DNA transcription and the translation of a number of proteins. The highlighted paragraph discusses how various JAK proteins express themselves differently in various cell types. The widespread expression of JAK1, JAK2, and TYK2 contrasts with the predominant presence of JAK3 in hematopoietic, myeloid, and lymphoid cells. The patent also lists a number of JAK2-binding PROTAC substances, such as ruxolitinib and baricitinib, which work by targeting JAK2 JH1 in people. In MHHCALL-4 cells, the substances were examined for their capacity to cause protein degradation, cytotoxicity, and effects on the JAK-STAT signaling pathway [120].

Protein kinases—which are often mutated in the human genome and involved in cellular processes such as cell apoptosis, signaling, signal transduction, and immune-response propagation—are linked to cancers. Both IL-1R and TLR dimerize as a result of ligand binding, and adaptor molecules are recruited to a highly conserved toll/IL-1R (TIR) domain on the cytoplasm. E3 ubiquitin ligases have received increased attention recently in drug development attempts because they are more desirable therapeutic targets. MDM2 inhibitors, which target the E3 ligase mouse double minute 2 homologue (MDM2), and von Hippel–Lindau (VHL) tumor suppressor, which is the substrate recognition component of the E3 ligase complex VCB, are two examples of specific ligands that have been developed to bind to these ligases. By responding to DNA damage or stress and controlling cell development, the tumor suppressor gene p53 also exerts a significant influence on apoptosis [121].

A possible method for causing tailored protein degradation is the proteolysis-targeting chimera (PROTAC) technology. In this strategy, heterobifunctional molecules are used to attract an E3 ubiquitin ligase to a particular protein of interest, causing the proteasome to degrade it. Indirectly affecting upstream signaling cascades, transcriptional programs, or epigenetic processes can result in PROTAC-mediated protein degradation. This technology has proven to be successful in numerous preclinical and clinical investigations, demonstrating its potential as a cutting-edge treatment strategy [122]. A promising small-molecule therapy approach for treating disorders connected to the androgen receptor (AR), such as prostate cancer, Kennedy’s disease, and cardiovascular conditions, is the PROTAC idea. The capacity of PROTACs to lower protein levels is one of its main benefits. When compared to conventional small-molecule inhibitors, this property allows for small-molecule degraders to achieve a more comprehensive target inhibition, which could lead to more effective treatments [123].

A prospective target for the therapy of numerous disorders, including cancer, is cyclin-dependent kinases (CDKs). Compound iCDK9, a highly selective CDK9 inhibitor, is one possibility. The possible toxicity of this inhibitor and our incomplete understanding of its mechanism, however, pose some limits. A class of bioactive molecules known as PROTAC (proteolysis-targeting chimeras) degraders can selectively stimulate the degradation of their target protein in vitro and in vivo, therefore lowering the dose-limiting toxicity of small-molecule medications [124]. Ao et al. (2023) recently developed and synthesized [124] bifunctional PROTAC compounds to target iCDK9 and show its hitherto unidentified target and pharmacological mechanism. The CD-5 chemical showed minimal toxicity in cells while selectively degrading CDK9. While CRBN-based PROTACs such as ARV-110 and ARV-471 have drawn a lot of attention for their therapeutic potential in treating cancer and other diseases, the study emphasizes that they have also been thoroughly investigated throughout the world and have been found to be effective in treating a variety of illnesses including viral infections, cardiovascular diseases, immune disorders, and neurodegenerative diseases [125].

Histone deacetylase 6 (HDAC6) may be a therapeutic target for the treatment of a number of disorders, according to a recent study. According to the study, since HDAC6 is essential for the activation of the NLRP3 inflammasome, targeting it may be useful in the treatment of inflammatory illnesses. In order to accomplish this, Cao Z et al. (2021) created an HDAC6 degrader with minimal cytotoxicity using the PROTAC approach, which targets proteolysis. They did this by combining pomalidomide, a CRBN E3 ligand, with a selective HDAC6 inhibitor produced from the natural product indirubin. Multiple cell lines, including active THP-1 cells, had their HDAC6 levels efficiently and arbitrarily decreased by the HDAC6 degrader [126]. Similarly to antibody–drug conjugates (ADCs), antibody–PROTAC conjugates have become a possible method for the selective administration of a broad-spectrum PROTAC to particular cell types. Due to their capacity to deliver cytotoxic drugs to cancer cells only, ADCs have become more and more prominent in the treatment of cancer [127].

He and colleagues created the aptamer–PROTAC conjugate (APC) in their 2021 study by affixing a BET-targeting PROTAC to the nucleic acid aptamer AS1411 (AS) with a cleavable linker. By improving the molecule’s (APR) capacity to target tumors in an MCF-7 xenograft model, this design also reduced toxicity while enhancing BET degradation and anti-tumor effects. The researchers used this method to improve the molecule’s distribution and selectivity, highlighting its potential for use in medicinal applications in the future [128].

Proteolysis-targeting chimeras (PROTACs) have become a promising approach for investigating pharmacological targets that are difficult to target with conventional methods. In recent years, these chimeric molecules have shown effective targeting capabilities, overcoming the challenges posed by traditional approaches. With conventional therapies, medication resistance is frequently unavoidable; nevertheless, this unique technique offers a means of overcoming it. In order to combat acquired drug resistance, PROTACs offer a potentially effective approach by promoting the targeted degradation of particular proteins.

## 11. Clinical Research on Proteolysis-Targeting Chimeras

The PROTACs currently undergoing clinical trials are included in Table 2 [129], and some of them, including ARV-110 and ARV-471, have shown encouraging results. The oral protein degradation drug ARV-110 selectively binds to AR and speeds up its breakdown [130]. Small-molecule medication ARV-110 has proven to be able to totally breakdown and remove AR (DC50 1 nM) in all examined cell lines. The oral treatment of ARV-110 at a dosage of 10 mg/kg, as reported in the reference [131], resulted in a considerable suppression of tumor growth in enzalutamide-insensitive tumors in a PDX model [132]. It was developed by Arvinas, Inc. to break down mutant androgen receptors, which are known to be clinically significant. This tiny chemical is called ARV-110. It has been shown that the ARV-110 can work well in a hyperandrogenic setting. Positive results from the first in-human phase I study of ARV-110 on patients with metastatic castrate-resistant prostate cancer (mCRPC) suggested its safety and tolerability. Early in January 2020, this information was released [133].

According to recently revealed interim clinical results from Arvinas, ARV-110 has demonstrated potential performance in a phase 1 clinical trial for patients with very-late-stage meta-static castration-resistant prostate cancer (mCRPC), with PSA reductions of >50% noted at doses greater than 280 mg; however, earlier research and numerous pre-treatments may cause tumor resistance to targeted androgen receptor (AR) therapy, which would reduce its effectiveness. A total of 84% of patients receiving ARV-110 treatment had non-AR gene mutations, according to a molecular biological examination of the patients. Despite the highly varied nature of the patient population in the phase 1 trial, Arvinas has found a subset of advanced patients with a molecular profile that suggests a particularly robust response to ARV-110.

## 12. Some Challenges of PROTAC Remain to Be Addressed

Because PROTAC molecules have a propensity to go outside of the traditional Lipinski “rule-of-five” area, developing them as possible therapeutic agents presents particular difficulties [134]. Because of this, there is still a lot to learn about how PROTAC penetrates the cell membrane, and further study is required to understand its uptake, dispersal, metabolism, excretion, and toxicity. Increasing cell absorption and bioavailability remains a critical problem to maintaining the required concentration of PROTAC for pharmacological efficacy. For these challenges to be overcome, molecules with optimal physicochemical properties are essential. The target protein FKBP12 is susceptible to protein degradation caused by the chemical RC32, which highlights the potential of improving PROTACs with suitable linkers to increase their permeability [135]. Due to these discoveries, the first oral AR PROTAC degrader for prostate cancer was created, and despite anticipated pharmacokinetic difficulties such as poor cell permeability and limited bioavailability, it has shown promise in clinical testing. Over 600 E3 ligases have been identified in the human body, and they all play a significant part in proteasome-mediated protein breakdown. Only a small portion of them, nevertheless, currently have useable small-molecule ligands [136]. One E3 ubiquitin ligase is used by the bulk of published PROTACs (proteolysis-targeting chimeras), which presents a significant obstacle to broadening the technology’s applicability. More specialized E3 ligases [137] and ligands that can be used in particular cells or tissues are being sought by researchers. A promising development for PROTACs has recently been made with the discovery of a ligand that attracts the arylhydrocarbon receptor (AhR) E3 ligase [138]. Only when PROTACs establish a sturdy ternary complex with a PROTAC-E3 ubiquitin ligase are they able to selectively ubiquitinate target proteins. The research of PROTAC-mediated ternary complexes, however, is currently limited by the difficulty of detecting and trapping the ternary crystal structures [139].

Additionally, the design of the linker in PROTACs is essential because PEG or linear alkyl chain architectures are vulnerable to oxidative metabolism, which reduces the concentration and duration of drug exposure and speeds up excretion from the body. There is not yet a set design rule that specifies how long and what kind of material the middle linker in PROTACs should be. Finally, medicinal chemists face a significant hurdle due to the intricacy of PROTAC molecules [140].

Designing an effective PROTAC molecule can be difficult since it depends on a number of elements, such as the choice of the linking site, the length of the connector, and the make-up of the conjugation vector. The “click reaction platform”, which provides important advantages such as higher throughput, scalability of synthesis, and increasing the space of chemical libraries, is one way to approach this difficulty systematically. It has long been difficult to create medications with exact design and specificity to control a variety of biological activities that target endogenous proteins. However, it is anticipated that in the near future, researchers from academia and the pharmaceutical sector will be able to jointly handle these difficulties in a satisfactory manner.

## 13. Conclusions and Perspective

Drug development is a growing field. Furthermore, the PROTAC innovation shows considerable promise as a novel method that provides advantage of a distinct MOA. The reports discussed here lend credence to the concept that the event-driven display put on by PROTAC has a number of advantages over the MOA’s occupancy-driven exhibit. Significant improvements from the previous year include improved binding and degradation-selectivity profiles towards homologous protein families and the ability to get through common inhibition-derived resistance mechanisms.

Additionally, it appears that the PROTAC’s technique can modify the locations where the inhibitor MOA fails. PROTACs have been developed as cutting-edge therapeutic approaches in drug discoveries throughout the past 20 years. More than ten PROTACs are now undergoing clinical studies. Event-driven PROTACs have the ability to exploit pharmaceutical targets that were previously inaccessible to traditional occupancy-driven inhibitors. The creation and deployment of PROTACs must overcome a number of challenges, such as a consistent negative impact from PROTAC’s operations outside the sick area. Additionally, Lipinski’s rule of five is frequently broken by the design of PROTACs, which might complicate their absorption, metabolism, distribution, and excretion.

In this investigation, we give a comprehensive overview of the strategies currently being used to improve the efficacy and selectivity of PROTACs. While certain PROTACs require additional biological investigations, such as in vivo research, safety profiles, and DMPK profiles, others need further optimization to improve their metabolic stability and potency. Since it can take some time, we predict that PROTACs will eventually reach the clinic. In order to better understand the biology of proteins before they even reach the clinic, resources such as the PROTACs, which relate to the functions for which small-molecule inhibitors have not yet been identified, are helpful. In many human diseases, PROTACs have a lot of potential. The next generation of PROTACs needs to be developed thoroughly in order to have greater selectivity, better pharmacokinetics, higher therapeutic efficacy, and less toxicity. It is crucial to create PROTACs as strong biological instruments to investigate the physiological and pathological functions of enzymes as well as their potential as cutting-edge therapeutic agents.

Ultimately, PROTACs that can be ingested would be a significant step in turning these molecules from a notion into a medication. The development of PROTACs into potent medicines and understanding their therapeutic potential are top research priorities.

## Figures and Tables

**Figure 1 molecules-28-04014-f001:**
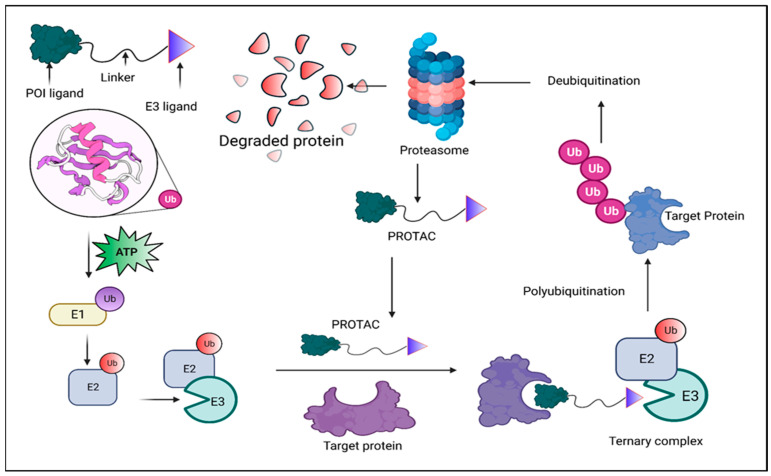
Proteolysis-targeting chimeras (PROTACs) are shown schematically. The PROTAC molecule is a heterobifunctional chemical that includes an E3 ligase recruiter and a ligand for the protein of interest (POI) (in blue). To initiate ubiquitination (seen in light blue) and subsequent proteasomal destruction, PROTACs force the E3 ligase and POI to be close to one another. PROTACs, which assist the transfer of ubiquitin, connect the POI and E3 ligase. Proteasomes identify the POI associated with ubiquitin and break it down into peptides. E3 ligase and the target protein can both be bound by the PROTAC molecule at the same time. Under the control of E1, E2, and E3 ligases, ubiquitins can be constantly transported to the target protein, leading to the polyubiquitination of the target protein. The target protein that has been polyubiquitinated is then added.

**Figure 2 molecules-28-04014-f002:**
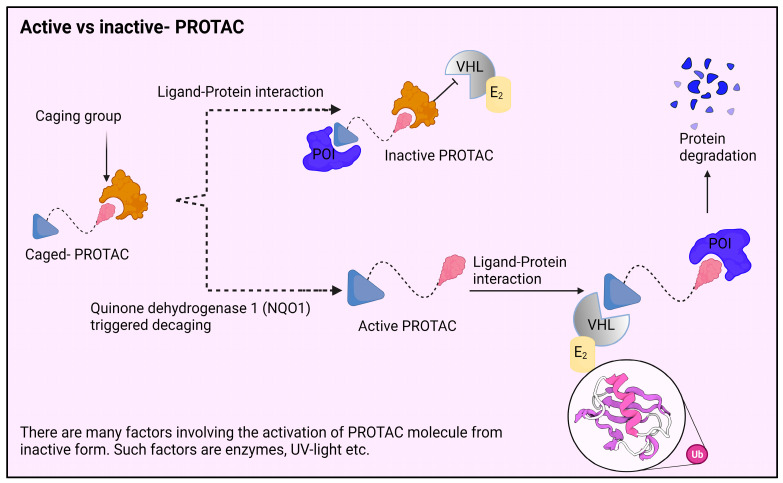
The components of PROTACs are tiny compounds that consist of a ligand that binds to the target protein, a ligand that binds to an E3 ubiquitin ligase, and a linker that joins the two ligands. The cellular ubiquitin–proteasome system can destroy particular proteins using the proteolysis-targeting chimera (PROTAC) technology, a chemical method for protein knockdown. The E3 ubiquitin ligase is brought into close proximity to the target protein by the PROTAC, which encourages the transfer of ubiquitin molecules onto the protein, marking it for degradation by the proteasome. Through the targeted degradation of proteins important in tumor growth, this strategy has demonstrated potential for cancer therapy. Nevertheless, the structure of PROTACs can affect their activity, and recent studies have looked at the use of enzyme-catalyzed activation to regulate PROTAC activity and improve its selectivity for the target protein.

**Figure 3 molecules-28-04014-f003:**
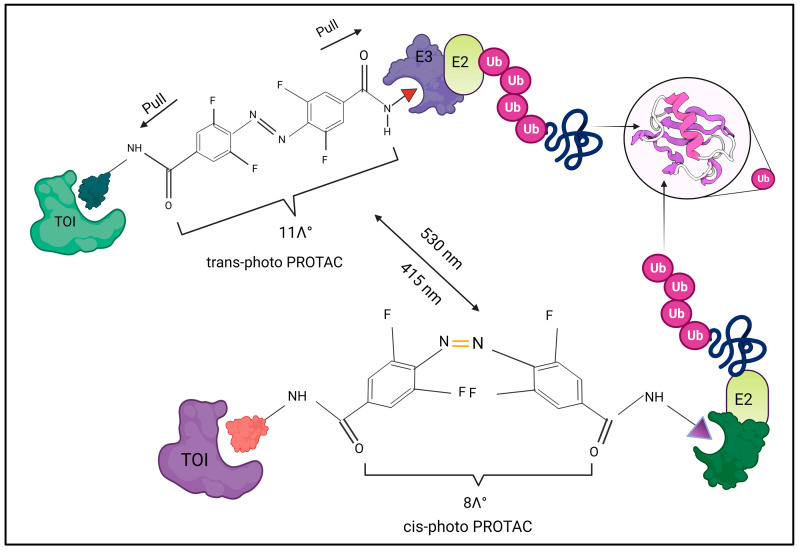
A photo-switchable BET PROTAC’s design philosophy. By substituting o-F4-azobenzene for the oligoether linker in ARV-771, an isomeric photo–PROTAC pair is produced, with the cis-isomer being much shorter than the trans-isomer.

**Figure 4 molecules-28-04014-f004:**
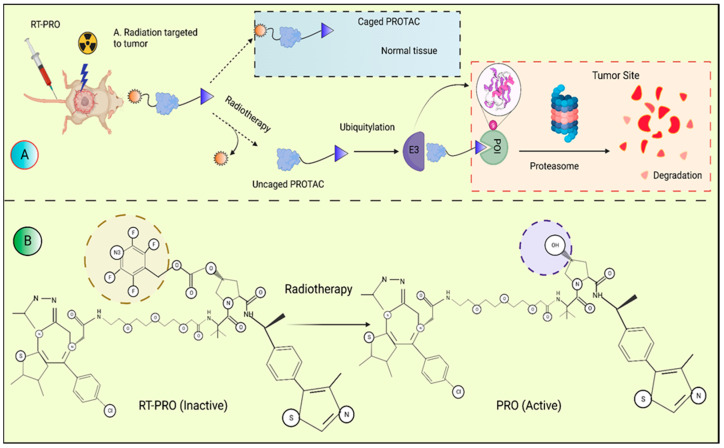
(**A**) A schematic illustration of the activation of RT-PRO in mice carrying tumors. In the absence of radiation, RT-PRO is inactive and has no harmful effects on healthy tissues. The prodrug is activated to release the PROTAC molecule PRO, while the tumor location is exposed to X-ray radiation. The target proteins are specifically destroyed by the released PRO at the tumor site under the influence of UPS. (**B**) The response of RT-PRO to PRO’s release. The radioactive PROTAC controls protein breakdown spatiotemporally and has synergistic anticancer effectiveness.

**Figure 5 molecules-28-04014-f005:**
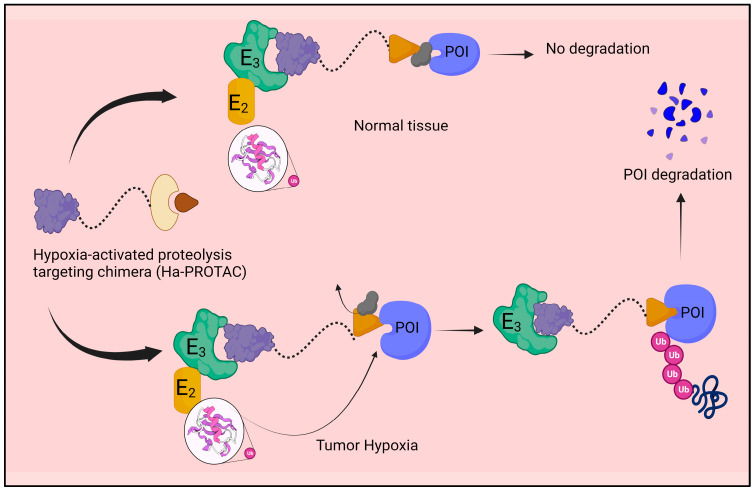
The hypoxia-activated proteolysis-targeting chimera (ha-PROTAC) is a novel group of small molecules designed to degrade proteins selectively in hypoxic conditions. It comprises three components: a ligand that binds to the target protein and a hypoxia-activated leaving group.

**Figure 6 molecules-28-04014-f006:**
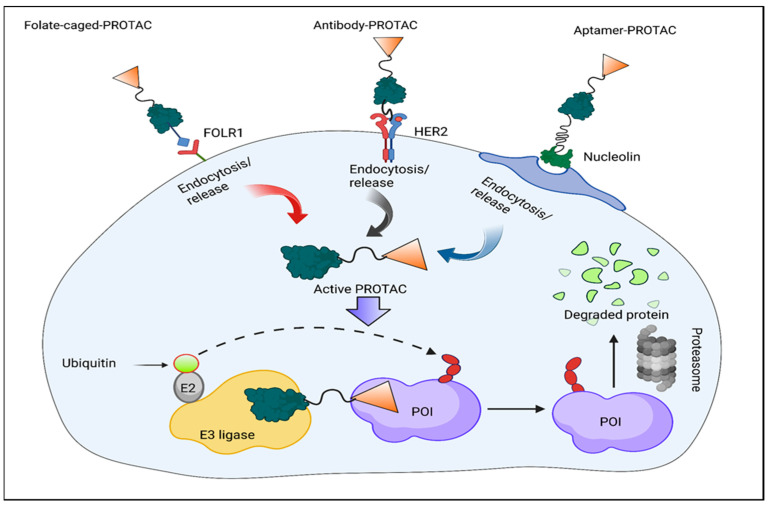
Schematic presentation of the design strategies for folate-caged PROTACs, antibody–PROTAC conjugates (Ab-PROTACs), and aptamer–PROTAC conjugates (APCs). Upon special recognition by the cell membrane receptor (e.g., FOLR1, HER2, and nucleolin), these PROTAC-based conjugates are taken up by cells via endocytosis. Then, the linker is cleaved by hydrolases to release the active PROTAC molecule, leading to target protein degradation.

**Figure 7 molecules-28-04014-f007:**
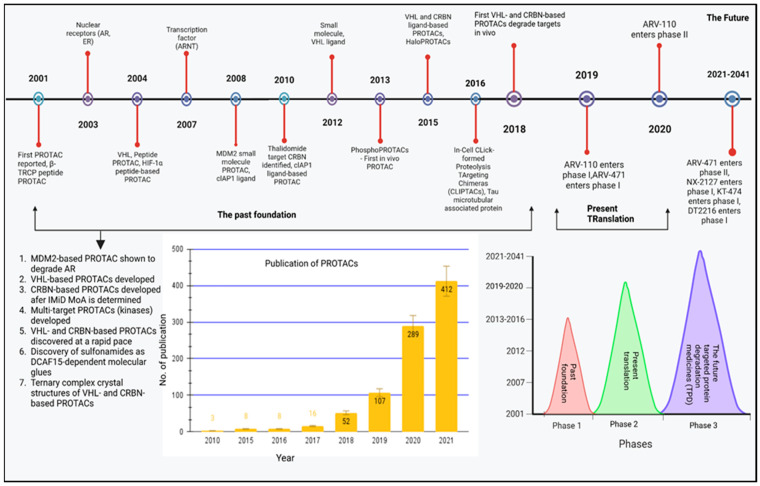
Timetable describing the evolution of PROTACs (2001–2016) and publication numbers during several years [56].

**Figure 8 molecules-28-04014-f008:**
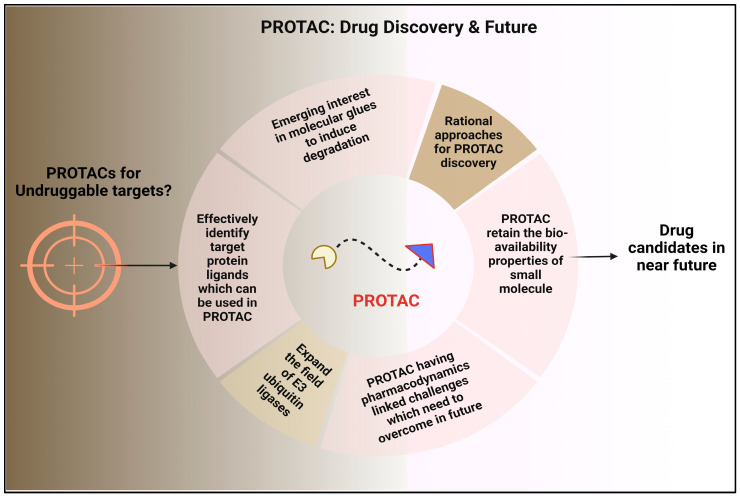
The Future of Drug Discovery with PROTAC Technology.

**Figure 9 molecules-28-04014-f009:**
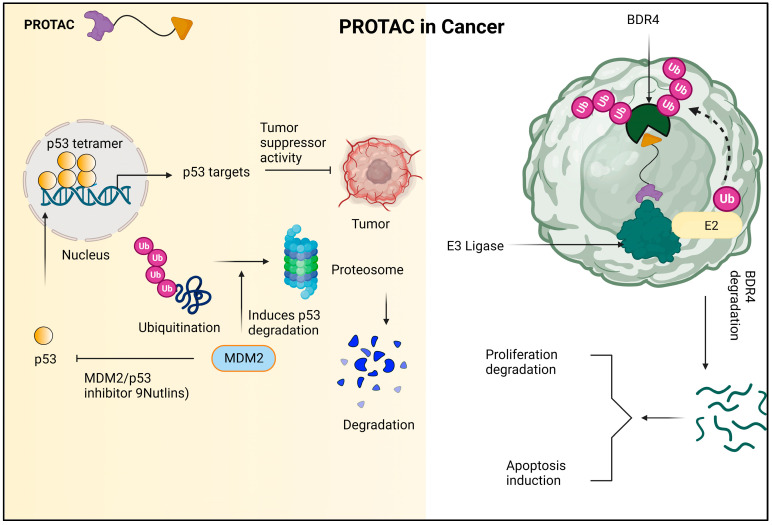
Due to the encouraging the ubiquitination and subsequent proteasomal destruction of the tumor suppressor protein p53, MDM2 has been identified as a possible target for anti-cancer therapy. This procedure is interfered with by nutlins and other MDM2/p53 interaction inhibitors, which prevent p53 from being degraded and instead cause it to accumulate. In turn, this strengthens p53′s anti-tumor properties and offers a viable plan for the creation of novel cancer treatments. The PROTAC molecule known as dBET1 is made up of two parts: thalidomide, which attracts the E3 ubiquitin ligase CRL4 CRBN, and JQ1, which binds to the oncoprotein BRD4. Using the cellular apparatus of the cell, dBET1 is designed to target BRD4 for breakdown by the proteasome. 7.2. PROTACs in neurodegenerative diseases.

**Figure 10 molecules-28-04014-f010:**
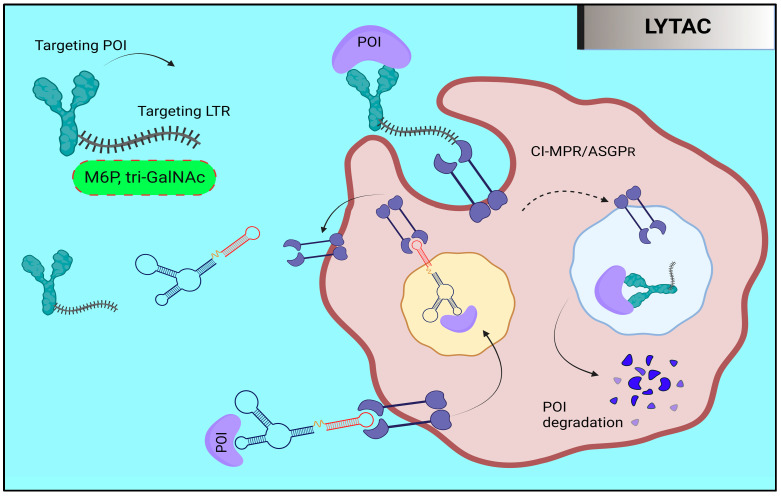
A tiny molecule or antibody attached to a ligand, known as a LYTAC (lysosomal-targeting chimera), interacts with lysosome-targeting receptors (LTRs) such as the CI-MPR and ASGPR to cause protein degradation. All human tissues include CI-MPR, whereas ASGPR is exclusively found in the liver. As a result, ASGPR-based LYTAC could target particular protein breakdown in the liver, whereas CI-MPR is present in all human tissues. Endocytosis is used to pick up the proteins of interest (POI) and LYTAC molecules after CI-MPR or ASGPR binds to them. In contrast, CI-MPR or ASGPR is recycled to the plasma membrane for use in the future, while the POI is later broken down by lysosomes.

**Figure 11 molecules-28-04014-f011:**
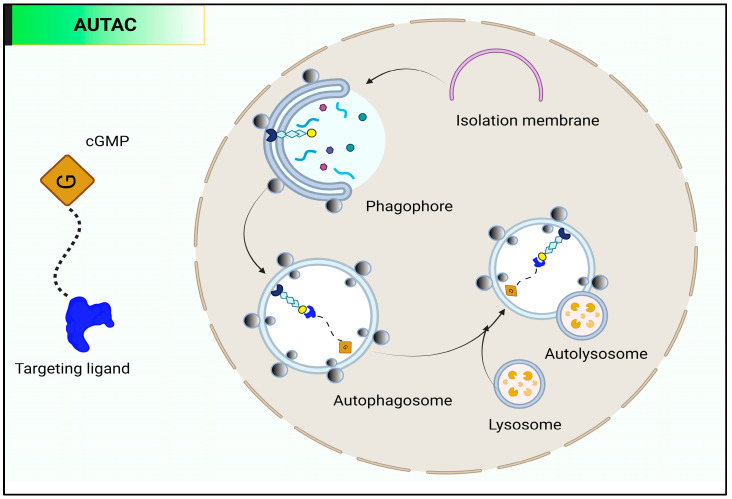
Specifically targeted proteins can be degraded by autophagy using a sort of synthetic chemical called AUTAC. Three components make up these molecules: a warhead that targets a particular protein of interest (POI), a linker, and a degradation tag based on cyclic guanosine monophosphate (cGMP). The POI is degraded by this tag by enlisting autophagosomes, cellular organelles, and structures that digest cytoplasmic proteins. The development of aberrant or undesirable proteins in the body can lead to a variety of disorders, and AUTACs have shown promise as potential treatments.

**Table 1 molecules-28-04014-t001:** PROTACs in Clinical Trials by the End of 2021.

Name of PROTAC	Target	Purpose	E3 Ligase	Target Receptor	Target Site	Name of PROTAC	Target	Purpose
ARV-110, Phase III	AR	Prostate cancer	CRBN	Nuclear receptor	Cytosol, nucleus	ARV-110, Phase III	AR	Prostate cancer
ARV-471, Phase II	ER	Breast cancer	CRBN	Nuclear receptor	Cytosol, nucleus	ARV-471, Phase II	ER	Breast cancer
AC682, Phase I	ER	Breast cancer	CRBN	Nuclear receptor	Cytosol, nucleus	AC682, Phase I	ER	Breast cancer
ARV-766, Phase I	AR	Prostate cancer	-	Nuclear receptor	Cytosol, nucleus	ARV-766, Phase I	AR	Prostate cancer
CC-94676, Phase I	AR	Prostate cancer	CRBN	Nuclear receptor	Cytosol, nucleus	CC-94676, Phase I	AR	Prostate cancer
DT2216, Phase I	BCL-XL	Liquid and solid tumors	VHL	Anti-Apoptotic protein	Cytosol, nucleus, MOM	DT2216, Phase I	BCL-XL	Liquid and solid tumors
FHD-609, Phase I	BRD9	Synovial sarcoma	-	Nuclear factor	Nucleus	FHD-609, Phase I	BRD9	Synovial sarcoma
KT-474, Phase I	IRAK4	Autoimmune diseases	-	Kinase	Cytosol	KT-474, Phase I	IRAK4	Autoimmune diseases
KT-413, Phase I	IRAK4	Diffuse large B cell lymphoma	CRBN	Kinase	Cytosol	KT-413, Phase I	IRAK4	Diffuse large B cell lymphoma
KT-333, Phase I	STAT3	Liquid and solid tumors	-	Nuclear factor	Cytosol, nucleus	KT-333, Phase I	STAT3	Liquid and solid tumors
NX-2127, Phase I	BTK	B cell malignancies	CRBN	Tyrosine kinase	Cytosol	NX-2127, Phase I	BTK	B cell malignancies
NX-5948, Phase I	BTK	B cell malignancies and Autoimmune diseases	CRBN	Tyrosine kinase	Cytosol	NX-5948, Phase I	BTK	B cell malignancies and Autoimmune diseases
CFT8634, IND-e	BRD9	Synovial sarcoma	CRBN	Nuclear factor	Nucleus	CFT8634, IND-e	BRD9	Synovial sarcoma
CFT8919, IND-e	EGFR L858R	NSLC	CRBN	Cell surface receptor	Cell membrane	CFT8919, IND-e	EGFR L858R	NSLC
CG001419, IND-e	TRK	Cancer and other indications	CRBN	Tyrosine kinase	Cell membrane	CG001419, IND-e	TRK	Cancer and other indications

**Table 2 molecules-28-04014-t002:** PROTACs under clinical trial investigation.

Therapeutics	Clinical Trial Number and Phase	Target	Toxicity Outcomes	Primary Efficacy Data
ARV110	NCT03888612, Phase 2	Prostate cancer	ARV-110 has an acceptable safety profile; however, co-administration of rosuvastatin with ARV-110 could produce toxic side effects.	The paragraph describes the results of a study or clinical trial involving a group of patients. Specifically, it states that out of 15 patients who received a 140 mg dose, 2 of them experienced a reduction in PSA (prostate-specific antigen) levels of more than 50%. Additionally, out of five patients who had either T878 or H875 mutations in AR (and received the same dose), two of them also had PSA reductions of over 50%. Finally, among the 15 patients who had wild-type AR, 2 of them (13%) had PSA reductions of over 50%.
ARV471	NCT04072952, Phase 2	Breast cancer	ARV-471 has been found to be well tolerated across all tested dosage levels with no reported treatment-related adverse events of grade 3 or 4. Additionally, no dose-limiting toxicities (DLTs) were reported during testing. These adverse events include nausea, which occurs in 24% of cases, arthralgia and fatigue, which occur in 19% of cases each, and decreased appetite, which occurs in 14% of cases.	In the ARV471 trial, which involved 21 adult patients, 1 patient achieved a confirmed partial response (PR) with a 51% reduction in the size of the target lesion. Two other patients had PRs, but they were not confirmed. Additionally, one patient had a stable disease with a reduction in target lesion size of over 50%. Overall, 42% of the 12 patients evaluated for clinical benefit response (CBR) achieved CBR.
KT474	NCT04772885, Phase 1	Autoimmune including AD, HS and RA	Not reported	Not reported
NX2127	NCT04830137, Phase 1	B cell malignancies	Not reported	Not reported
ARV-471, Drug: Ribociclib	NCT05573555, Phase 2	Breast Cancer	Less toxicities in ARV-471 in combination with Ribociclib	Overall Survival
Drug: ARV-471Drug: Abemaciclib	NCT05548127, Phase 1, Phase 2	Breast Cancer	The number of participants with dose-limiting toxicities; dose-limiting toxicity rate for ARV-471 in combination with Abemaciclib.	% of participants gaining clinical advantage enhances

## Data Availability

Not applicable.

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
