# Peer review of "PROTACs: Emerging Targeted Protein Degradation Approaches for Advanced Druggable Strategies"

_molecules, 2023, doi:10.3390/molecules28104014_

Round 1
Reviewer 1 Report
The authors delivered a great summary of the PROTACs, especially on the controllable PROTACs. The authors also have a very thorough summary of the application of PROTACs in potential therapeutics. However, there are many aspects the authors can improve.
1. The English needs rework significantly. In many places, the grammar and wording needs to be reconsidered. For example, the authors used "protein destruction" instead of "protein degradation". In many places, it is hard to understand what the authors are trying to deliver.
2. The overall structure of the review has no focus. In comparison of other PROTACs review, the authors may put emphasis on controllable PROTACs, the therapeutic applications of PROTACs and their challenges.
3. There are very few figures in the paper. It is really helpful to make figure illustration especially in the section 5 to 7.
4. Some references are missing in the paper. For example, "There are numerous different protein categories, like nuclear receptors, kinases, G protein-coupled receptors (GPCRs), transmembrane proteins, small GTPases, epigenetic protein, transcription factor and protein aggregates, have benefited from the application of the PROTAC technology over time." needs references.
Author Response
Reviewer 1
The authors delivered a great summary of the PROTACs, especially on the controllable PROTACs. The authors also have a very thorough summary of the application of PROTACs in potential therapeutics. However, there are many aspects the authors can improve.
Comment 1: The English needs rework significantly. In many places, the grammar and wording needs to be reconsidered. For example, the authors used "protein destruction" instead of "protein degradation". In many places, it is hard to understand what the authors are trying to deliver.
Response 1. We improved the English language and removed grammatical errors. The revised version is now improved.
Comment 2: The overall structure of the review has no focus. In comparison of other PROTACs review, the authors may put emphasis on controllable PROTACs, the therapeutic applications of PROTACs and their challenges.
Response 2. We have added the therapeutic application and challenges of PROTAC in therapeutic application field.
Comment 3: There are very few figures in the paper. It is really helpful to make figure illustration especially in the section 5 to 7.
Response 3. Added new figures in the mentioned sections for better understanding
Comment 4: Some references are missing in the paper. For example, "There are numerous different protein categories, like nuclear receptors, kinases, G protein-coupled receptors (GPCRs), transmembrane proteins, small GTPases, epigenetic protein, transcription factor and protein aggregates, have benefited from the application of the PROTAC technology over time." needs references.
Response 4. We added references according to the mentioned suggestion. Thank you for pointing it out.
…………………………….Thank you for your valuable suggestion……………………………

Reviewer 2 Report
This is a timely and well-written review article that provides a comprehensive summary of current knowledge and recent advances in developing PROTACs as a precision degradation approach to retard tumorigenesis. The article is clearly written, however, the concerns listed below should be addressed before its publication.
1. The images for figures are blurry, it will be nice for authors to use high-resolution images.
2. It will be nice for the authors to devote a figure panel to the concept of making PROTAC controllable (inactive versus active) before jumping into the very complicated design of all the platform of stimuli-induced PROTACs.
3. It will nice to devote one figure to illustrate the mechanism of action for hypoxia induced PROTAC
4. It will be nice for the authors to also briefly introduce LYTAC and AUTAC.
Author Response
#Reviewer 2
This is a timely and well-written review article that provides a comprehensive summary of current knowledge and recent advances in developing PROTACs as a precision degradation approach to retard tumorigenesis. The article is clearly written, however, the concerns listed below should be addressed before its publication.
Comment 1: The images for figures are blurry; it will be nice for authors to use high-resolution images.
Response 1. In the revised paper all the new figures are now in 600 dpi. The blur figure has been removed.
Comment 2: It will be nice for the authors to devote a figure panel to the concept of making PROTAC controllable (inactive versus active) before jumping into the very complicated design of all the platform of stimuli-induced PROTACs.
Response 2. Figure added (Fig 2) which depicts the clear mechanism of active and inactive PROTAC molecule under certain condition (enzymes, light).
Comment 3: It will nice to devote one figure to illustrate the mechanism of action for hypoxia induced PROTAC
Response 3. New figure has been added (Figure 4).
Comment 4: It will be nice for the authors to also briefly introduce LYTAC and AUTAC.
Response 4. We have added overview on LYTAC, AUTAC with figures and also added additional information including GluTAC and AbTAC. Thank you for this valuable and informative suggestion.
…………………………….Thank you for your valuable suggestion…………………………

Round 2
Reviewer 1 Report
I am satisfied with the modifications the authors made. English is greatly improved with many more figure illustration. I have no further comments.